# Efficacy and Safety of Vasopressin and Terlipressin in Preterm Neonates: A Systematic Review

**DOI:** 10.3390/ijerph192113760

**Published:** 2022-10-22

**Authors:** Abdulrahman Al-Saadi, Katelyn Sushko, Vivian Bui, John van den Anker, Abdul Razak, Samira Samiee-Zafarghandy

**Affiliations:** 1Division of Neonatology, Department of Pediatrics, Sultan Qaboos University, Muscat 123, Oman; 2Faculty of Health Sciences, School of Nursing, McMaster University, Hamilton, ON L8S 4L8, Canada; 3Department of Pharmacy, Hamilton Health Sciences, Hamilton, ON L8L 2X2, Canada; 4Pediatric Pharmacology and Pharmacometrics, University Children’s Hospital Basel (UKBB), University of Basel, 4055 Basel, Switzerland; 5Division of Clinical Pharmacology, Children’s National Hospital, Washington, DC 20010, USA; 6Intensive Care and Department of Pediatric Surgery, Erasmus MC Sophia Children’s Hospital, 3000 CB Rotterdam, The Netherlands; 7Division of Neonatology, Department of Pediatrics, King Abdullah bin Abdulaziz University Hospital, Princess Norah Bint Abdulrahman University, Riyadh 11564, Saudi Arabia; 8Department of Pediatrics, Monash University, Melbourne 3800, Australia; 9Division of Neonatology, Department of Pediatrics, McMaster University, Hamilton, ON L8S 4L8, Canada

**Keywords:** preterm neonate, arginine vasopressin, terlipressin, hypotension

## Abstract

Introduction: The use of arginine vasopressin (AVP) and terlipressin to treat hypotension in preterm neonates is increasing. Our aim was to review the available evidence on the efficacy and safety of AVP and terlipressin for use in preterm neonates. Methods: MEDLINE, EMBASE, the Cochrane Central Register of Controlled Trials, Web of Science, and Google Scholar from inception to September 2021 were searched for studies of AVP and terlipressin in the treatment of hypotension of any cause in preterm neonates. Primary outcomes were improvement in end-organ perfusion and mortality. The risk of bias assessment and certainty of the evidence were performed using appropriate tools. Results: Fifteen studies describing the use of AVP (n = 12) or terlipressin (n = 3) among 148 preterm neonates were included. Certainly, the available evidence for the primary outcome of end-organ perfusion rated as very low. AVP or terlipressin were used to treat 144 and 4 neonates, respectively. Improvement in markers of end-organ perfusion was reported in 143 (99%) neonates treated with AVP and 3 (75%) treated with terlipressin. The mortality rate was 41% (n = 59) and 50% (n = 2) for neonates who received AVP and terlipressin, respectively. Hyponatremia was the most frequently reported adverse event (n = 37, 25%). Conclusion: AVP and terlipressin may improve measured blood pressure values and possibly end-organ perfusion among neonates with refractory hypotension. However, the efficacy–safety balance of these drugs should be assessed on an individual basis and as per the underlying cause. Studies on the optimal dosing, efficacy, and safety of AVP and terlipressin in preterm neonates with variable underlying conditions are critically needed.

## 1. Introduction

Arginine vasopressin (AVP), also known as antidiuretic hormone, is a cyclic-nonapeptide hormone that has a role in osmotic and cardiovascular hemostasis [1]. AVP mediates its multimodal function mainly through three receptors, V1-vascular (V1R), V2-renal (V2R), and V3-pituitary (V3R) [2]. The vasoconstrictor effect of AVP is mediated by activating V1Rs and stimulating tissue-specific G-protein-coupled receptors (GPCRs), found in high concentrations in vascular smooth muscles [1,3,4]. However, AVP can also cause vasodilation by releasing nitric oxide in the vascular endothelium of the skeletal muscles, skin, and some visceral organs [3,4,5]. The V2Rs, located on the distal tubules of the kidneys, are mainly known for mediating the antidiuretic effects of AVP. However, activation of V2Rs may stimulate rapid release of platelets and coagulants from bone marrow and vascular endothelium [3]. The V3Rs mediate the release of adrenocorticotropic hormone (ACTH) from the pituitary gland and are also involved in a variety of AVP’s neurobehavioral modulatory actions [3,6].

Terlipressin is a non-selective synthetic long-acting analogue of vasopressin [7,8]. Terlipressin demonstrates comparable pharmacodynamic effects to AVP. However, it is more V1R selective [6,9,10,11]. The pharmacokinetic properties of the two analogues are also different. AVP has a shorter half-life than terlipressin (six minutes versus six hours, respectively) [6,12]. The longer half-life of terlipressin results in an extended elimination time of approximately 50 min, possibly limiting its use in the neonatal population compared to AVP, the effects of which fade within 20 min [7,8,9].

Fluid-refractory catecholamine-resistant shock, particularly septic shock, is one of the leading causes of morbidity and mortality in the neonatal intensive care unit, with devastating consequences in preterm neonates [4,12,13]. The goal of management is to restore systemic circulation with appropriate use of fluid therapy and increase vascular tone with vasopressors [14]. However, in shock that is resistant to catecholamines, the lack of response to vasopressors is a significant challenge.

Currently available data suggest that in refractory vasodilatory shock, concentrations of endogenous vasopressin remain low. This knowledge, along with the strong vasoconstrictor properties of AVP [3,4,15,16], has provided the biologic plausibility to support the use of AVP and its analogues in a state of refractory shock [17]. Current data suggest an increasing trend in the use of AVP and terlipressin among preterm neonates. This increase is despite a lack of data on the optimal dosing regimen and the short- and long-term safety of AVP in preterm neonates [18,19,20].

The aim of our systematic review was to review the available evidence on the efficacy and safety of AVP and terlipressin in preterm neonates with hypotension of any cause.

## 2. Methods

### 2.1. Study Design

The reporting of this systematic review followed the Preferred Reporting Items for Systematic Reviews and Meta-Analyses (PRISMA) checklist [21]. The protocol for this review was prospectively registered (PROSPERO, registration number CRD42021236094) and published [22]. Box 1 provides the Population, Intervention, Control, Outcome (PICO) framework used for this review.

Box 1PICO framework.**Population:** Preterm neonates born at less than 37 weeks’ gestation with hypotension (defined as mean blood pressure less than gestational age or hypotension requiring fluid or vasoactive therapy) or persistent pulmonary hypertension.**Intervention:** Arginine vasopressin or terlipressin administered intravenously, initiated at any time, and for any duration as a primary or rescue treatment for hypotension or persistent pulmonary hypertension. **Comparator:** Standard treatment, placebo, or any other vasoactive agent. **Outcomes:** Primary Outcomes: (i) improvement in end-organ perfusion defined as an increase in systolic, diastolic or mean blood pressure, or an increase in urine output, a decrease in the need for inotropes, or a reduction in serum lactate as reported by the authors of the primary studies, (ii) mortality prior to discharge. Secondary Outcomes: (i) major neurosensory disability defined as moderate to severe motor or cognitive impairment or severe visual or hearing impairment as reported by the authors of the primary studies; and (ii) the occurrence of adverse events defined as peripheral tissue ischemia, gastrointestinal events (occurrence of perforation, necrotizing enterocolitis, or gastrointestinal bleed), hepatic events, renal events or hyponatremia as reported by the authors of the primary studies.

### 2.2. Data Sources

Ovid MEDLINE (1964–September 2021), EMBASE (1974–September 2021), Web of Science (1900–September 2021), and the Cochrane Central Register of Controlled Trials (CENTRAL) were systematically searched. This search strategy, containing database-specific subject headings and text word terms for concepts, was first developed in MEDLINE (Ovid interface) and was translated as appropriate for the other databases (Figure 1). We also searched the bibliographies of relevant studies and citations of the studies included for additional references. Using Google Scholar, we searched for relevant studies that were not commercially published, such as conference abstracts, dissertations, policy documents, and book chapters. No language or study design limitations were applied. We excluded animal studies and duplicate studies. A professional librarian peer-reviewed our strategy using the Peer Review for Electronic Search Strategies (PRESS) guideline [23] (Appendix A).

### 2.3. Study Selection

All original research studies and conference abstracts describing AVP or terlipressin as primary or rescue treatment for hypotension of any cause in preterm neonates born at less than or equal to 37 weeks gestational age were eligible. These included randomized controlled trials (RCTs), quasi-RCTs, prospective and retrospective cohort studies, descriptive studies, case series, and case reports. We included studies with mixed populations (term and preterm neonates) if separate data for preterm neonates were available. Studies were eligible for inclusion irrespective of the dose, administration frequency, and duration of AVP or terlipressin treatment. Standard practice or other therapeutic interventions were the comparators in studies with a control group. If there was no comparator group, the reported efficacy and safety of AVP/terlipressin were extracted.

Our primary outcomes were: (1) improvement in end-organ perfusion, defined as an increase in systolic, diastolic, or mean blood pressure, or an increase in urine output, a decrease in the need for inotropes, or a reduction in serum lactate, as reported by the authors in the primary studies and (2) mortality before discharge. Our secondary outcomes were: (1) major neurosensory disability, defined as a moderate to severe motor or cognitive impairment or a severe visual or hearing impairment, as reported by the authors of the primary studies and (2) occurrence of adverse events, defined as peripheral tissue ischemia, gastrointestinal events, hepatic events, renal events, or hyponatremia, as reported by the authors in the primary studies (Appendix B).

We used Covidence as the primary screening and data extraction tool. Two independent reviewers (AA, KS) screened the titles and abstracts of retrieved studies to assess their eligibility. The eligible studies were then reviewed in duplicate at the full-text level by the same reviewers. We resolved disagreements through discussion with a third reviewer (SSZ).

### 2.4. Data Extraction and Synthesis

Two reviewers (AA, KS) independently conducted data extraction from the full-text studies meeting the inclusion criteria using a standardized data extraction form developed in Covidence (Appendix C). We resolved any disagreements throughout the data extraction process through discussion with a third reviewer (SSZ). We conducted a narrative synthesis of the study results structured around the Population, Intervention, Comparator, Outcome (PICO) framework. We described the details of the population (gestational weeks at birth, birth weight, postnatal age, indication for treatment), intervention, comparator, and outcome.

### 2.5. Risk of Bias Assessment

Two independent reviewers (AA, KS) conducted a qualitative assessment of included studies. The Tool for Evaluating the Methodological Quality of Case Reports and Case Series [24], the adapted Retrospectoscope for Reducing Bias in Chart Review Studies [25] and the Cochrane Collaboration Risk of Bias 2.0 tool [26] were used, as appropriate. The scales that we proposed in our original protocol [22] were not applicable to the included studies, thus, we modified the scales for our risk of bias assessment based on applicability.

### 2.6. Assessment of the Certainty of Evidence

Two reviewers (AA, KS) rated the certainty of the evidence using the Cochrane Grading of Recommendations Assessment, Development, and Evaluation approach (GRADE) [27]. We resolved any disagreements through discussion with a third reviewer (SSZ).

### 2.7. Data Analysis

The included articles did not present comparative effect estimates. Therefore, we did not conduct a meta-analysis.

## 3. Results

We identified 3780 articles as a result of the database and grey literature searches. After deduplication, we screened 2678 papers at the title and abstract and 53 at the full-text level (Figure 2). We included 15 studies in the final review [18,28,29,30,31,32,33,34,35,36,37,38,39,40,41].

### 3.1. Characteristics of Studies and Population

Among the 15 included studies, the most common type of study was case report (n = 6, 33%) [35,36,37,38,40], followed by case series (n = 4, 26%) [28,32,33,34]. The remaining studies included three (20%) retrospective descriptive studies [29,30,31], one (7%) abstract of a case report [41], and one (7%) randomized controlled trial [18] (Table 1). None of the included articles presented comparative effect estimate statistics.

The 15 studies described 223 episodes of AVP or terlipressin therapy in 148 preterm neonates. AVP and terlipressin were administered to 144 and 4 neonates in the treatment of 219 (98%) and 4 (2%) episodes of hypotension of variable underlying conditions, respectively. Of all neonates, 141 (95%) were extremely or very preterm (<32 weeks). The majority of episodes and treatment initiation times (n = 216, 97%) were within the first two weeks of life. The most common underlying causes of hypotension were refractory shock due to variable aetiologies (i.e., congenital cardiac or surgical, n = 63, 28%), refractory septic shock (n = 57, 26%), persistent pulmonary hypertension (n = 36, 16%), and early transient hypotension (n = 10, 4%).

### 3.2. Treatment Details

#### 3.2.1. AVP

Twelve studies described the use of AVP for the treatment of hypotension in 144 neonates (219 episodes) [18,28,29,30,31,32,33,34,35,38,39,41], with median (minimum, maximum) gestational age (GA) and postnatal age (PNA) of 26 (23, 36) days, respectively. AVP was a rescue treatment in 10 studies (134 neonates, 199 episodes, 93%). One RCT examined the effect of AVP versus dopamine in the initial treatment of early transient hypotension (n = 10, 7%) [18]. If adequate mean BP was not reached after the highest study drug dose, one dose of intravenous hydrocortisone (1 mg/kg) was administered. The treatment groups were comparable in baseline characteristics, except for neonates in the vasopressin group that had a lower mean PaCO2 (*p* < 0.05) and bicarbonate (17.5 vs. 19.5, *p*-value not significant) and higher base deficit (10.3 vs. 8.9, *p*-value not significant) during study drug administration and received fewer doses of surfactant (*p* < 0.05; Table 2). Furthermore, seven neonates in the vasopressin and three in the dopamine treatment group received a normal saline bolus (10 mL/kg) before the study drug initiation. The study reported successful treatment, defined as reaching mean blood pressure 2 mmHg above gestational age (in weeks), for nine subjects in each treatment group, with three and one neonates in the vasopressin and dopamine groups also requiring hydrocortisone to achieve target blood pressure. Urine output was similar in the vasopressin and dopamine treatment groups (3.5 ± 1.4 vs. 4.4 ± 1.4), and there was no report of other markers of end organ perfusion. Mortality was reported as a secondary outcome and analyzed along with BPD as a composite outcome. Four and two neonates in the vasopressin and dopamine treatment group died, but composite outcome for BPD or death were similar in both groups (*p* = 0.26). In one case report, AVP was the initial pharmacotherapy for hypotension secondary to hypertrophic obstructive cardiomyopathy [39].

A continuous infusion of AVP was administered in all cases. However, two studies provided additional bolus doses [27,40] for two neonates (2 episodes). The majority of studies (n = 10, 83%) titrated the dosing of the AVP infusion, with a mean (minimum, maximum) starting dose and maximum doses of 0.029 (0.001, 0.14) and 0.13 (0.018, 0.48) mcg/kg/h, respectively.

#### 3.2.2. Terlipressin

Three case reports described the use of terlipressin for the treatment of refractory hypotension as a rescue therapy in four preterm neonates (four episodes) with a median (minimum, maximum) GA and PNA of 30 (25, 34) weeks and 7.5 (4, 11) days, respectively. Terlipressin was administered as a bolus with cumulative dosing of 0.12 and 0.2 mg/kg/day in two studies (three neonates, three episodes) and as a bolus and infusion of 5 mcg/kg followed by 1–10 mcg/kg/h in one study (one neonate, one episodes) [37].

### 3.3. Outcomes

#### 3.3.1. End-Organ Perfusion

##### Vasopressin

Overall, 11 of the included studies (92%) with a total of 143 (99%) neonates and 219 (99%) episodes reporting improvement in end-organ perfusion, with 11 studies (142 (98%) neonates and 217 (99%) episodes) reporting increased blood pressure numbers, 3 studies (51 (35%) neonates and 126 (57%) episodes) reporting decreased ionotropic use, 7 studies (83 (57%) neonates and 92 (42%) episodes) reporting improved urine output, and 2 studies (32 (22%) neonates and 65 (29%) episodes) reporting decreased serum lactate.

##### Terlipressin

Three of the included studies (100%) with a total of three (75%) neonates (3 episodes) reported improvement in end-organ perfusion. In two (50%) of the cases, the increase in blood pressure was reported along with decreased inotropic use. None of the available studies reported an improvement in urine output or a decrease in serum lactate levels after terlipressin administration.

#### 3.3.2. Mortality, Major Neurosensory Disability, and Adverse events

##### Vasopressin

Of all neonates treated with AVP, 59 (41%) died. Examining the rate of mortality as per the underlying conditions, neonates with gastrointestinal (5 of 8, 68%) and cardiac disease 2 of 3, 66%) as the causes of refractory hypotension had the highest mortality rate.

Two studies reported the neurodevelopmental outcomes of 10 survivors [33,38]. Severe neurodevelopmental disability was reported in two neonates (20%) receiving AVP infusion for refractory hypotension secondary to PPHN [33]. Of the 12 studies (144 neonates, 85%) that reported adverse events, they occurred in 52 (36%) preterm neonates [18,28,29,30,31,33,35,38,39]. The most frequent adverse event was hyponatremia (39 episodes in 37 neonates, 25%) [18,30,31,33,35,39], which was severe in 8 episodes (21%) [31,35]. Other reported adverse events were increased serum creatinine value (one episode, one neonate) [29], gastric perforation (one episode, one neonate) [38], elevated liver enzymes (five episodes, five neonates), decreased platelet count (one episode, one neonate), severe mitral regurgitation (two episodes, two neonates) [29] and hepatic necrosis on autopsy (one episode, one neonate) [28].

##### Terlipressin

Two of four neonates (50%) treated with terlipressin for refractory septic and vasodilatory shock of unknown origin died. Normal neurodevelopmental assessments were reported for the remaining two survivors treated for refractory septic shock and PPHN [36,37]. There were no reports of adverse events among neonates treated with terlipressin.

### 3.4. Risk of Bias Assessment

We assessed the risk of bias of the included retrospective studies by answering the questions, adapted from the Retrospectoscope for Reducing Bias in Chart Review Studies [25]. All four studies did not provide information regarding six out of 10 required questions. Thus, the lack of sufficient data limited our ability to judge the risk of bias (Table 2).

The included case series, case reports, and case studies were evaluated across four domains: risk of selection bias, ascertainment bias, causality bias, and reporting bias, using the Tool to Evaluate the Methodological Quality of Case Reports and Case Series [24]. Selection and causality were the domains through which bias might have been introduced in most studies (Table 3).

We assessed the risk of bias for the included RCT using the Cochrane Risk of Bias 2.0 tool [26]. We judged the RCT to have a high risk of bias. The risks arising from the randomization process, missing outcome data and measurement of the outcome was low, while we judged the risk of bias from outcome selection and deviation from the intended intervention to be high (Figure 3).

### 3.5. Assessment of the Certainty of the Evidence

As we did not conduct a meta-analysis, we followed the recommendations of Murad et al. for rating the certainty of evidence in the absence of a single estimate of effect [41]. The certainty of the evidence for the primary outcome end-organ perfusion was very low (Table 4). The risk of bias assessment for the RCT determined that the study had a high risk of bias. Furthermore, there was no information available about several risks of bias domains for the retrospective studies. Therefore, we judged the included studies to have serious methodological limitations. The participants, interventions/exposures, and comparators were directly comparable to our clinical question. We did not suspect indirectness. For imprecision, the total number of participants in the included studies did not meet the threshold of 400 (n = 148) [42]. Furthermore, none of the included studies reported a 95% confidence interval. Thus, we judged concerns about imprecision as serious. For inconsistency, the results showed an unclear direction of effect of AVP/terlipressin on end-organ perfusion (improved blood pressure in 92% of studies; decreased need for inotropes in 83% of studies; improved urine output in 50% of studies; decreased serum lactate in 29% of studies). We judged concerns about inconsistency as serious. We did not strongly suspect publication bias as both negative and positive studies were included in our review and our search strategy was comprehensive.

## 4. Discussion

The results of this systematic review show that AVP may improve blood pressure and urine output among neonates with refractory hypotension of various aetiologies. The observed increase in urine output among neonates receiving AVP is supported by the hypothesis that in a state of septic shock where endogenous vasopressin is low, vasopressin increases the glomerular filtration rate of the kidneys [43]. Interestingly, in the four neonates receiving terlipressin, a synthetic analogue of AVP with targeted selectivity for V1 receptors, there was no report of increase in urine output. This finding, along with the long half-life of terlipressin raises concern for its use in preterm neonates with highly dynamic needs.

In the only available study (RCT) describing the use of vasopressin in preterm neonates with early transient refractory hypotension [18], although vasopressin increased blood pressure, no other outcomes related to the end-organ perfusion were reported. Furthermore, the higher number of neonates in the vasopressin group needing hydrocortisone treatment (three vs. one) raises question on efficacy of vasopressin. Investigators reported significantly lower CO_2_ levels and less need for surfactant in neonates treated with vasopressin, as an indicator of beneficial effects of vasopressin on pulmonary hemodynamics. However, these neonates also had a lower bicarbonate (17.5 vs. 19.5) and higher base excess (10.3 vs. 8.9), which could indicate that the lower CO_2_ was in compensation for the more severe metabolic acidosis. If this lower level of bicarbonate was an adverse outcome associated with vasopressin treatment is another important question in need of investigation. The mortality rate during the treatment period was similar in the two groups (one vs. one) but three more neonates in vasopressin vs. one in the dopamine treatment group died prior to discharge. The cause of death for these neonates is not reported. Considering this lack of data, in preterm neonates with hypotension in the immediate postnatal transitional period, where myocardial maladaptation, uncompensated decrease in preload and delayed adrenal recovery are the main known associated pathophysiology [44,45,46], the use of vasopressin cannot be recommended.

We examined mortality as per the underlying cause of refractory hypotension and found it to be consistently high (30–70%) among all different groups. As this population has a high background rate of morbidity and mortality already, distinguishing the possible adverse effects of vasopressin in these poorly controlled studies is impossible. Information on the neurodevelopmental outcome of these neonates was scarce. Data were available on only ten survivors (12%) in the AVP group and two (100%) in the terlipressin group. Although severe neurodevelopmental disability occurred in only two neonates receiving AVP (20%), the lack of methodological precision in the included studies precludes any interpretation of the findings.

Outcome assessment for the occurrences of adverse events was presented in 12 of the 14 studies on AVP and all studies on terlipressin, which included 144 and four neonates, respectively. Hyponatremia occurred in 25% of cases, with most episodes reported as non-severe (n = 31, 79%). Gastric perforation and hepatic necrosis were the only other serious adverse events that occurred (n = 2, 1%). Potent pressor action of vasopressin on hepatosplenic circulation and other microcirculatory blood flow is the main area of concern for its use in preterm neonates. Juvenile animal data have been conflicting with studies in septic models, showing a marked redistribution of portal, pancreatic, and even renal blood flow despite an increase in urine output [47]. This decrease in microcirculatory blood flow to the hepatic, pancreatic, upper gastrointestinal tract, and renal systems has been reported in up to 30% of studied species, suggesting the fast and sustained increase in blood pressure following administration of AVP or its analogues might not be associated with an increase in systemic blood flow. On the contrary, two studies on the effect of AVP in juvenile pigs receiving cardiopulmonary bypass or having mesenteric ischemia showed that this drug can preserve capillary density and tissue blood flow and limit endothelin expression as a marker of intestinal microcirculatory disturbance [48,49]. These conflicting results, although limited to animal models, raises caution in use of vasopressin in preterm neonates and emphasizes the urgent need for population pharmacokinetic-pharmacodynamic data. The use of these drugs should also be limited to randomized controlled trials, specifically in patients with underlying conditions, such as early transient hypotension, where the scarce available data do not support a favorable efficacy–safety balance.

## 5. Limitations

We attempted to collect all available literature on the use of AVP and terlipressin in the treatment of hypotension among preterm neonates. However, the existing evidence mainly consisted of studies at the lowest level of medical evidence. Thus, we must interpret the results cautiously, as they are subject to vulnerabilities, including the inability to establish a cause–effect relationship between exposures and outcomes. Only one of the included studies used an RCT design, but this study also remained limited in the small sample size and limited reports of important outcomes.

## 6. Conclusions

Our review suggests that AVP and terlipressin may improve measured blood pressure values and hemodynamic indices among neonates with refractory hypotension. However, clinicians need to assess the efficacy–safety balance when using these drugs in each preterm neonate with refractory hypotension, individually. The limited data obtained through the current systematic review is cautiously supportive of the use of AVP in preterm neonates with refractory septic shock. However, close monitoring of indices of end organ perfusion, including renal, hepatic, pancreatic and gastrointestinal is highly warranted. There is no biologic plausibility, dosing, efficacy, or safety data to support the use of AVP in neonates with early transient hypotension. As AVP appears to be a viable option in critically ill preterm neonates with refractory hypotension, studies on its optimal dosing, efficacy, and safety profile in preterm neonates with variable underlying conditions is critically needed.

## Figures and Tables

**Figure 1 ijerph-19-13760-f001:**
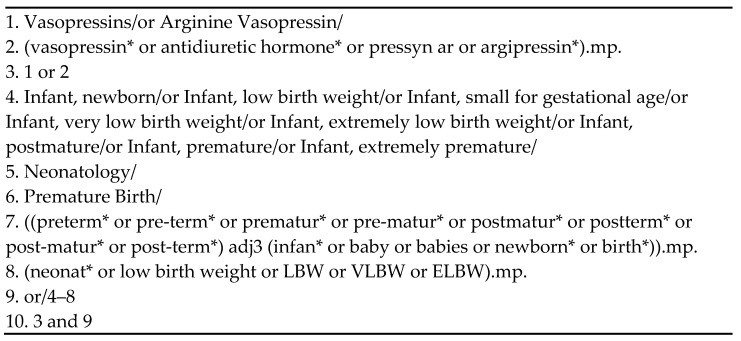
Search strategy for Ovid MEDLINE.

**Figure 2 ijerph-19-13760-f002:**
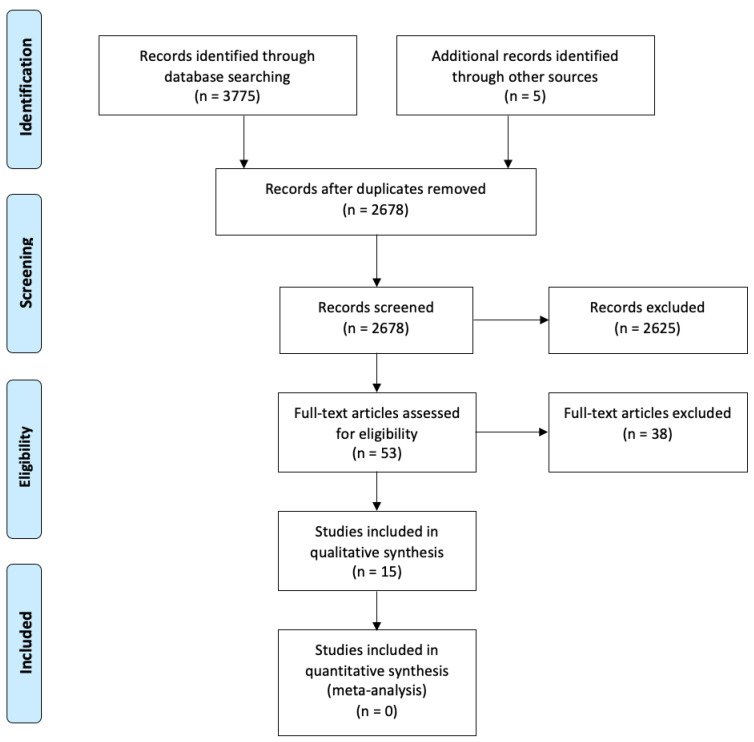
PRISMA flow diagram of study selection process.

**Figure 3 ijerph-19-13760-f003:**
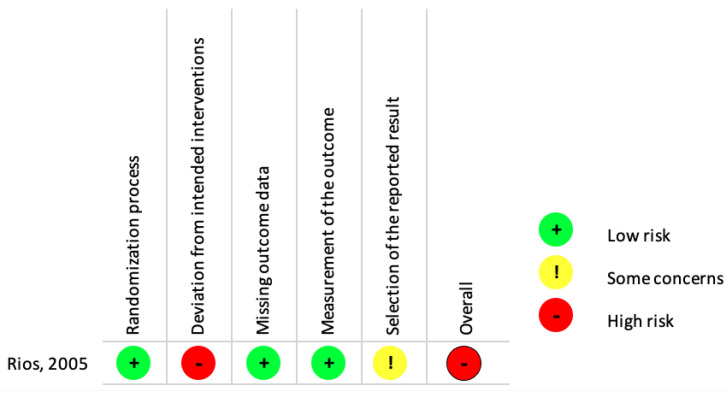
Risk of bias summary-Review of authors’ judgements about each risk of bias item for the included randomized controlled trial [18].

**Table 1 ijerph-19-13760-t001:** Characteristics of included studies on arginine vasopressin (AVP) and terlipressin on hypotension and persistent pulmonary hypertension.

	Methods	Population				Intervention						Comparator			Outcomes
Author, Year	Design	Sample Size	GA (wks)	BW (gr)	PNA at Intervention (days)	Indication	Treatment Details					Comparator Details		Study Period (months)	Outcomes
							No. Pts. (episodes)	Initial Dose (AVP u//kg/h) (Terlipressin mcg/kg)	Infusion/ Bolus	Max Dose (μ/kg/h)	Concurrent Medications	No. Pts. (Episodes)	Treatment		
**AVP**															
Rios, 2015 [18]	Double Blind RCT	70	25.6 (1.4)	705 (154)	0.25 (0.01)	Early hypotension	10	0.01	Infusion	0.04	Hydrocortisone (rescue)	10	Dopamine infusion 5–20 mcg/kg/min	24	↑ SBP: NR ↑ DBP: NR ↑ MBP: 9 vs10 ↑ UO: NS ↓ Inotropic Support: NR ↓ Lactate: NR Death: 4 vs. 2 Deaths &BPD: 8 vs. 10 Neuro disability: NR AE: Hyponatremia: NS
Meyer, 2006 [28]	Case Series	6	24.5–27.6	Septic shock: 600 (30) AKI: 770 (110)	NR	Shock-Sepsis 3 AKI 3	6	Sepsis: 0.035 AKI: 0.01	Infusion (1 bolus & infusion)	0.36	Norepinephrine Epinephrine Enoximone	N/A	N/A	21	↑ SBP: NR ↑ DBP: NR ↑ MBP: 6/6 ↑ U/O: 6 ↓ Inotropic Support: NR ↓ Lactate: 2 Deaths: 4 Neuro disability: NR AE: 1 hepatic necrosis on autopsy
Ni, 2017 [29]	Retrospective Study	39	29.1	2220 (1275)	NR	Septic shock 5 NR 34	39	0.014		0.032	Dopamine Epinephrine Dobutamine Hydrocortisone	N/A	N/A	47	↑ SBP: (*p* < 0.0005) ↑ DBP: (*p* < 0.01) ↑ MBP: Yes (*p* < 0.001) ↑ UO: Yes (*p* < 0.05) ↓ Inotropic Support: NR ↓ Lactate: NS Deaths: 26 Neuro disability: NR AE: 1 ↑ creatinine
Ikegami, 2010 [30]	Retrospective Study	22	24.9 (1.4)	658 (142)	11 {0–34}	Shock-Sepsis: 8 Adrenal dysfunction: 5 GI perforation: 3 Cardiogenic shock: 1 CNS disturbance: 3 Unclassified: 2	22	0.001		0.02	Dopamine Dobutamine Hydrocortisone	N/A	N/A	47	↑ SBP: 18/22 (*p* < 0.0001) ↑ DBP: 18/22 (*p* < 0.0001) ↑ MBP: NR ↑ U/O: 18/22 (*p* < 0.0001) ↓ Inotropic Support: NR ↓ Lactate: NR Deaths: 7/22 Neuro disability: NR AE: 5 ↑ liver enzymes 2 mitral regurgitations 3 ↓ platelet 6 hyponatremia
Budniok, 2020 [31]	Retrospective Study	26	26 [24–29]	800 [615–1274]	14 (4–25)	Shock- Sepsis: 16 Shock-PPHN: 11 Hypertrophic cardiomyopathy: 1	26 (33)	0.018		0.039	Hydrocortisone	N/A	N/A	60	↑ SBP: (*p* < 0.001) ↑ DBP: (*p* < 0.001); ↑ MBP: (*p* < 0.001); U/O: (*p* = 0.06) ↓ Inotropic Support: (*p* < 0.001) ↓ Lactate: (*p* = 0.005) Deaths: 4 Neuro disability: NR AE: 28 severe hyponatremias
Bidegain, 2010 [32]	Case Series	20	25 {23–27}	680 {400–980}	10 {1–240}	Shock- Sepsis: 28 NEC: 5	20 (33)	0.01		0.08	Dopamine Epinephrine Hydrocortisone	N/A	N/A	31	↑ SBP: NR ↑ DBP: NR ↑ MBP: (*p* < 0.002); ↑ U/O: No ↓ Inotropic Support: (*p* < 0.05) ↓ Lactate: NS Deaths: 13/20 Neuro disability: NR AE: NR
Mohamed, 2020 [33]	Case Series	13	31.4 (3.3)	1762 (590)	1.2 (0.15)	Shock-PPHN	13	0.006		0.018	INO Milrinone	N/A	N/A	54	↑ SBP: (*p* < 0.05) ↑ DBP: No ↑ MBP: No ↑ U/O: NS ↓ Inotropic Support: NR ↓ Lactate: NR Deaths: 5 Neuro disability: 2 AE: Hyponatremia
Kaga, 2013 [34]	Case Series	4	23.0 (22.5–23.5)	466 (414–563)	24 (2–31)	Refractory hypotension	4 (9)	0.018		0.48	Dopamine Hydrocortisone Dexamethasone	N/A	N/A	15	↑ SBP: (*p* = 0.03) ↑ DBP: (*p* = 0.01) ↑ MBP: (*p* = 0.01) ↑ U/O: (*p* = 0.01) ↓ Inotropic Support: NR ↓ Lactate: NS Deaths: None Neuro disability: NR AE: None
Leister, 2020 [35]	Case Report	1	32.0	NR	NR	Shock-PPHN	1	0.14		0.14	Milrinone Hydrocortisone iNO	N/A	N/A	N/A	↑ SBP: NR ↑ DBP: NR ↑ MBP: NR ↑ U/O: NR ↓ Inotropic Support: NR ↓ Lactate: NR Deaths: None Neuro disability: NR AE: Severe hyponatremia
Ruf, 2018 [38]	Case Report	1	34.0	2515	1.5	Refractory hypotension	1	0.06		0.06	Dobutamine Noradrenalin Epinephrine Hydrocortisone	N/A	N/A	N/A	↑ SBP: Yes ↑ DBP: Yes ↑ MBP: Yes ↑ UO: Yes ↓ Inotropic Support: NR ↓ Lactate: NR Deaths: None Neuro disability: None AE: Gastric perforation
Bhatia, 2010 [41]	Case Report Abstract	1	29.0	1690	NR	Refractory hypotension	1	NR	AVP; Infusion vs. bolus NR	NR	Dopamine Dobutamine Hydrocortisone	N/A	N/A	N/A	↑ SBP: NR ↑ DBP: NR ↑ MBP: NR ↑ U/O: NR ↓ Inotropic Support: Yes ↓ Lactate: NR Deaths: None Neuro disability: NR AE: None
Boyd, 2020 [39]	Case report	1	36.0	4630	9.2	Hypotension-hypertrophic obstructive cardiomyopathy	1	0.018		Initial dose ↑ by 0.006–0.012	Alprostadil iNO	N/A	N/A	N/A	↑ SBP: (*p* = 0.028) ↑ DBP: (*p* = 0.009) ↑ MBP: (*p* = 0.004) ↑ UO: No ↓ Inotropic Support: NR ↓ Lactate: NS Deaths: None Neuro disability: None AE: Hyponatremia
**Terlipressin**															
Lopez-Suarez, 2009 [36]	Case Report	2	28.0 34.0	780 1660	NR 11	Shock-Sepsis	2	0.02 mg/k every 4 h 0.02 mg/kg every 6 h	Terlipressin Bolus	0.02 mg/k every 4 h 0.02 mg/kg every 6 h	Dopamine Dobutamine Norepinephrine	N/A	N/A	N/A	↑ SBP: NR ↑ DBP: NR ↑ MBP: 1 ↑ U/O: 1 ↓ Inotropic Support: 2 ↓ Lactate: NR Deaths: 1 Neuro disability: None AE: None
Bissolo, 2012 [40]	Case Report	1	25.3	920	14	Shock-Sepsis	1	50 q 6 h	Terlipressin bolus	50 q 6 h	Dopamine Dobutamine Epinephrine Noradrenaline Hydrocortisone	N/A	N/A	N/A	↑ SBP: No ↑ DBP: No ↑ MBP: No ↑ U/O: No ↓ Inotropic Support: No ↓ Lactate: No Deaths: 1 Neuro disability: NA AE: None
Oulego-Erroz, 2020 [37]	Case Report	1	33.0	2010	4	Shock-PPHN	1	Bolus: 5 Infusion: 1 mcg/kg/h	Terlipressin bolus and infusion	10 mcg/kg/h	Dopamine Dobutamine Norepinephrine Epinephrine hydrocortisone Bosentan	N/A	N/A	N/A	↑ SBP: NR ↑ DBP: NR ↑ MBP: Yes ↑ U/O: NR ↓ Inotropic Support: Yes ↓ Lactate: NR Deaths: None Neuro disability: None AE: None

**Table 2 ijerph-19-13760-t002:** Risk of bias assessment of individual retrospective studies.

Adapted Retrospectoscope for Reducing Bias in Chart Review Studies										
First Author, Year	Chart Review an Appropriate Method to Answer Question?	Investigator Conflict of Interest or Bias?	Patient Sample Representative?	Needed Variables in the Records?	Chart Abstraction Systematic?	Missing or Conflicting Data?	Abstractor Blinding Bias?	Abstractors Sufficiently Trained?	Abstractors Sufficiently Monitored?	Chart Abstraction Reliable?
Ni, 2017 [29]	Y	PN	PY	PN	NI	NI	NI	NI	NI	NI
Ikagemi, 2010 [30]	PY	PN	PY	PN	NI	NI	NI	NI	NI	NI
Budniok, 2020 [31]	PY	PN	PY	PN	NI	NI	NI	NI	NI	NI

Y, yes; PY, probably yes; N, no; PN, probably no; NI, no information.

**Table 3 ijerph-19-13760-t003:** Risk of bias assessment of individual case series, case reports, and case studies.

Tool for Evaluating the Methodological Quality of Case Reports and Case Series: Domains and Abbreviated Leading Explanatory Questions							
	Selection Bias	Ascertainment Bias	Causality				Quality of Reporting/ Description
First Author	Is the Patient(s) Representative of the Investigating Centre?	Was Exposure/Outcome Adequately Ascertained?	Was Alternative Cause Ruled Out?	Was There a Challenge/Rechallenge Phenomenon?	Was There a Dose-Response Effect?	Was Follow-up Long Enough?	Was the Case(s) Described in Sufficient Detail?
Meyer, 2006 [28]	Yes	Yes/Yes	No	No	No	Yes	Yes
Bidegain, 2010 [32]	Yes	Yes/Yes	No	No	No	Yes	Yes
Mohamed, 2020 [33]	Yes	Yes/Yes	No	No	Yes	Yes	Yes
Kaga, 2013 [34]	Unclear	Yes/Yes	Yes	No	No	Yes	Yes
Leister, 2020 [35]	Unclear	Yes/Yes	No	No	Yes	Yes	Yes
Lopez-Suarez, 2009, [36]	Unclear	Yes/Unclear	No	No	No	Yes	Yes
Oulego-Erroz, 2020, [37]	Unclear	Yes/Yes	No	No	No	Yes	Yes
Ruf, 2018 [38]	Yes	Yes/Yes	No	No	No	Yes	Yes
Boyd, 2020 [39]	Yes	Yes/Yes	No	No	No	Yes	Yes
Bissolo, 2012 [40]	Unclear	Yes/Yes	No	No	No	Yes	Yes
Bhatia, 2010 [41]	Unclear	Yes/Yes	Unclear	No	No	Unclear	No

**Table 4 ijerph-19-13760-t004:** Summary of findings.

AVP/Terlipressin Compared to the Standard Treatment Approach for Hypotension and Persistent Pulmonary Hypotension in Preterm Neonates			
Outcome	Effect	Number of Participants	Certainty in the Evidence
End-organ perfusion assessed using the following outcomes: systolic, diastolic or mean blood pressure, need for inotropic support, urine output, and serum lactate.	The direction of effect was unclear; the majority of studies showed improvements in blood pressure and need for inotropes but not urine output and serum lactate.	144 (1 randomized trial; 3 retrospective studies; 4 case series; 4 case reports; 1 case report abstract)	Very low ⊕OOO (due to serious risk of bias, imprecision and inconsistency)

## Data Availability

All data generated or analyzed during this study are included in this article. Further enquiries can be directed to the corresponding author.

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
