# Peer review of "Efficacy and Safety of Vasopressin and Terlipressin in Preterm Neonates: A Systematic Review"

_ijerph, 2022, doi:10.3390/ijerph192113760_

Round 1

Reviewer 1 Report

Dear Authors,

The presented systematic review of vasopressin and terlipressin in premature babies meets the requirements for this type of work. Summarizes information on these drugs, including limitations due to a lack of original research.

The Authors of the manuscript analyze the available literature in order to answer the question whether vasopressin is an effective and safe drug in premature babies suffering from hypotension caused by various reasons. Increased use of vasopresin and terlipressin is reported in premature infants with hypotension, which is explained by the low concentration of endogenous vasopressin in catecholamine-resistant shock with vasodilation. At the same time, there is a lack of conclusive data on optimal dosages and potential side effects.

Severe hypotension is still a problem of prematurity difficult to control, especially in the extreme premature babies and / or in the course of sepsis. Lack of a rapid return of blood pressure adequate for normal organ perfusion leads to serious complications, often irreversible. On the other hand, too rapid and excessive increase in blood pressure is a significant risk factor of intraventricular haemorrhage in this group of patients. However, the existing literature on this subject is very poor. For this reason, the topic is important and certainly requires further in-depth observations and research.

The article systematizes the data on the use of vasopressin and terlipressin in premature babies. Review articles on vasopressin in newborns, including premature babies, were published almost 10 years ago, and one - on all cardio- and vasoactive drugs used in 2018. A. Al-Saadi's work is comprehensive, focused on the population of premature babies and prepared in accordance with the requirements for this type of study. 

The methodology is based on the PICO scheme and the PRISMA checklist. The time period covered by the analysis is broad, including the most important databases. The method of conducting the analysis seems to ensure the objectivity of the assessment.

The conclusions refer to the purpose of the work, but their value is limited by the number of available publications on this topic, which is rather small. The Authors conclude that vasopressin and terlipressin should be considered in preterm infants with treatment-resistant shock, however a case-by-case assessment is needed to balance the benefits and risks for a particular patient. Undoubtedly, any doctor dealing with a severly ill premee and seaking the advice in the medical publications would expect unambiguousness authors’ opinions, clear guidelines regarding the indications for use, dose and duration of therapy. The relative scarcity of data does not allow for precise answers to these questions. However, it seems important to summarize the current medical experience with the use of this medication in a very sensitive patient population.

Author Response

Thank you for this comment.

Reviewer 2 Report

This is a systemic review of the use of vasopressin and terlipressin in premature infants. Totally 144 and 4 cases, respectively, from the literature, were reviewed. I will suggest removing the 4 terlipressin cases from the review since the number is too low, and its long half-life of 6 hours, makes it clinically unsuitable for premature infants. I do not agree with including the 36 cases of persistent pulmonary hypertension, which is aimed at reducing right-to-left shunt instead of true hypotension. Try to avoid repeating the same information, such as the content in Box 1 being the same as described in the upper part of page #4. Similar redundancy is seen throughout the manuscript. Although I have some concerns about this review, with a limited number of reports available, the content is worth sharing with clinicians. The authors have done a good in evaluating literature with objective tools to evaluate each report.

Reviewer 3 Report

I would like to thank the authors for an interesting and well written paper.

This systematic review on “Efficacy and Safety of Vasopressin and Terlipressin in Preterm

Neonates: A Systematic Review” is a welcome addition to the literature that is very deficient on this topic. The authors have clearly described the methods used and discussed the results of the systematic review considering current limitations. I fully agree with the authors on the limitations of the literature, and I believe these can only be addressed through multicenter prospective studies. The study adds important information in the field of neonatology, and I believe will spark an interest in further research on the risk and benefit of using vasopressin in preterm infants.

I have no concern with the study as it is well-design and cover all aspect of vasopressin use in preterm infants.

Author Response

Thank you for this comments.

Reviewer 4 Report

Manuscript "Efficacy and Safety of Vasopressin and Terlipressin in Preterm Neonates: A Systematic Review" by Abdulrahman Al-Saadi et al.

The systematic review covers a clinically highly significant topic. The manuscript summarizes available data about vasopressin/terlipressin treatment for preterm infants with hypotension.

Comment 1:

-        Lines 61-63: “…The goal of management is to restore circulation with aggressive fluid therapy…”. In my view, the reference provided is not appropriate. First, because “preterms” are not the focus of the update. Second, because “aggressive” fluid resuscitation is uncommon and often not the solution in the majority of preterms <32 weeks of gestation. As mentioned in the reference, “a more graded” approach reflects common practice. Please provide evidence for “aggressive” in preterm infants or rephrase the sentence.

Comment 2:

-        Lines 65-66: please provide reference for a study population of preterm infants or specify the study population for “currently available data”.

 Comment 3:

-        PICO framework – population: the authors define the population of preterm neonates born at less than 37 weeks’ gestation with hypotension. The majority of NICU use this definition based on gestational age. However, it is not clear why the authors use “or” instead of “and” when adding “hypotension requiring fluid or vasoactive therapy”. By using “or” one can infer that application of fluids without measuring a mean blood pressure below gestational age fulfils criteria. The same holds true for “or” – “persistent pulmonary hypertension”. Please specify and explain since common practice varies heavily between centres, e.g. depending on whether neonatal echocardiography is routinely performed, one can detect signs of persistent pulmonary hypertension even though mean blood pressure might still be above gestational age. Similarly, one might assume that decreased urine output reflects “hypotension” and prescribes fluids, although mean blood pressure might still be above mean blood pressure. In other words, did the authors include preterm infants without mean blood pressure below gestational age if fluids were applied or pulmonary hypertension diagnosed? Please comment.

 Comment 4:

-        PICO framework – outcomes and lines 108-110: assessing and defining end-organ perfusion is challenging. However, it not clear to me, why and based on which physiologic rationale systolic and diastolic blood pressures were listed. While associating mean blood pressure with perfusion is common in intensive care, systolic and diastolic pressures require justification. Please comment.

 Comment 5:

-        PICO framework – outcomes and lines 285-289 (discussion): I would like to challenge the view and assumption that increased urine output reflects improved end-organ perfusion. Increased urine output can be associated with a better perfusion, but is a rather weak surrogate marker, especially in the context of vasopressin. In addition, it is not specified how urine output is assessed: via bladder catheter, via weighing, via urine bag? One of the references provided, please see line 289, reference 42, originates from pediatric patients. Please comment and provide age-specific data supporting the assumption that increased urine output reflects better perfusion.

 Comment 6:

-        Line 174: “postnatal age of 26 weeks”. Please double check whether “days” are missing.

 Comment 7:

-        Lines 325-327 and lines 334-335: Please unequivocally rephrase the sentences where non-human and non-preterm data are referenced. The readers should effortlessly understand whether data originates from preterm/term neonates or pediatric patients, and especially whether data originates from adult or neonatal animals.

 Comment 8:

-        Line 355: please comment and explain why, how, and which pancreatic indices should be assessed.

Round 2

Reviewer 4 Report

The authors have addressed the issues raised by the reviewer.